# How Adipocytes Orchestrate Inflammation Within Adipose Tissue?

**DOI:** 10.3390/biom16010059

**Published:** 2025-12-30

**Authors:** Romane Higos, Gianluca Renzi, Paul Taillandier, Fatiha Merabtene, Christine Rouault, Jimon Boniface Abatan, Mélanie Lambert, Isabelle Dugail, Karine Clément, Geneviève Marcelin, Salwan Maqdasy, Christophe Breton, Simon Lecoutre

**Affiliations:** 1Nutrition and Obesities: Systemic Approach Research Group, Nutriomics, Sorbonne Université, Institut National de la Santé et de la Recherche Médicale (INSERM), 75013 Paris, France; romane.higos@inserm.fr (R.H.); paul.taillandier@etu.sorbonne-universite.fr (P.T.); fatiha.merabtene@inserm.fr (F.M.); christine.rouault@inserm.fr (C.R.); jimon.abatan@sorbonne-universite.fr (J.B.A.); isabelle.dugail@inserm.fr (I.D.); karine.clement@inserm.fr (K.C.); genevieve.marcelin@inserm.fr (G.M.); simon.lecoutre@inserm.fr (S.L.); 2Department of Medicine, Karolinska Institutet Hospital, 14186 Stockholm, Sweden; gianluca.renzi@ki.se; 3U1349 Institut National de la Santé et de la Recherche Médicale, 93022 Bobigny, France; melanie.lambert@univ-paris13.fr; 4Département de Génie Biologique, Institut Universitaire de Technologie (IUT), 93022 Bobigny, France; 5Department of Nutrition, Pitie-Salpêtriere Hospital, Assistance Publique-Hôpitaux de Paris, 75013 Paris, France; 6U1283-UMR8199-European Genomic Institute for Diabetes (EGID), Université de Lille, INSERM, Centre National de la Recherche Scientifique (CNRS), Centre Hospitalier Universitaire (CHU) Lille, Institut Pasteur de Lille, 59000 Lille, France

**Keywords:** adipocyte, obesity, fibro-inflammation

## Abstract

Adipose tissue is far more than a passive reservoir for surplus energy: it is an active metabolic and endocrine organ that senses nutrient availability and orchestrates systemic energy balance. When caloric intake chronically exceeds expenditure, adipocytes become engorged with lipids and exposed to metabolic, mechanical, and hypoxic stress. To adapt, they initiate a fibro-inflammatory response that may be protective in the short term. As this response becomes chronic, adipocytes lose their metabolic flexibility, acquire a maladaptive fibro-inflammatory phenotype, and contribute to the cascade of inflammation, insulin resistance, and metabolic disease that characterizes obesity. In this review, we dissect the cellular and molecular cues that trigger fibro-inflammation, from nutrient excess and mitochondrial stress to hypoxia and immunometabolic rewiring, and highlight how these processes reshape adipocyte identity and tissue homeostasis.

## 1. Introduction:

Formerly viewed as a passive, merely an inert reservoir of energy, white adipose tissue (WAT) has now been recognized as a metabolically and immunologically dynamic organ. Although adipocytes occupy 90–95% of tissue volume, they represent only ~20% of total cells. The remaining cells form the stroma vascular fraction (SVF) harboring a rich diversity of immune cells, accounting for up to 60% of SVF cells in obesity [1,2,3,4]. These include distinct populations of macrophage and monocyte subsets, B and T lymphocytes, neutrophils, eosinophils, basophils, mast cells, dendritic cells, natural killer cells, and innate lymphoid cells [2,3,5,6]. Each type occupies specialized anatomical niches that sustain their survival and shape their functional programming [5,7]. Adipocytes critically integrate environmental cues and orchestrate the local immune response through the active secretion of a broad repertoire of cytokines and hormones, collectively referred to as adipokines [8,9,10]. They also shed extracellular vesicles (EVs) that act as potent messengers across tissues and organs, shaping the inflammatory landscape far beyond adipose depots [9]. Interestingly, a recent study suggests that they are capable of antigen presentation, directly modulating T cell responses [11] (Figure 1).

This intimate crosstalk takes on pathological significance in obesity, as adipocytes become active drivers of inflammation. This, in turn, promotes the recruitment and activation of immune cells, pushing the tissue into a state of chronic, low-grade inflammation. This so-called “meta-inflammatory” state underpins adipocyte dysfunction and local insulin resistance, propagates systemic insulin resistance, and accelerates the development of type 2 diabetes mellitus (T2DM) [3,12]. A strong correlation between inflammatory burden and the degree of insulin resistance has positioned inflammation as a pivotal driver of metabolic disease [3,13,14,15,16,17,18]. Experimental models reinforce this concept: targeted depletion of pro-inflammatory immune subsets or blockade of chemokine and cytokine signaling consistently protects adipocyte function and improves insulin sensitivity [3,13,19,20,21,22]. Yet the primary trigger that ignites this inflammatory cascade remains poorly defined. Alterations in fatty acid homeostasis, extracellular matrix (ECM) remodeling, mechanical stress, hypoxia, oxidative and endoplasmic reticulum stress, mitochondrial dysfunction and associated quality control processes, as well as direct disruption of insulin signaling, have all been implicated [3,19,23] (Figure 1 and Figure 2). However, the causal hierarchy linking these events to metabolic decline remains unresolved [24].

Although adipose tissue inflammation is traditionally viewed as preceding adipocyte dysfunction, recent findings in both mice and humans indicate that it may instead arise after the onset of insulin resistance [21,25,26,27]. Moreover, inflammation has been observed to persist within WAT even after significant weight loss and improved glycemic control [28,29]. These observations raise the possibility that inflammation in metabolic tissues may not solely act as a trigger of insulin resistance, but may also participate in serving adaptive roles, preserving WAT plasticity, supporting extracellular matrix remodeling, promoting angiogenesis, and enabling adipogenesis [30,31,32]. Indeed, acute inflammatory signaling is essential for normal adipose tissue function, orchestrating tissue remodeling that allows excess lipids to be safely stored through the generation of new, healthy adipocytes. By contrast, failure of adipose tissue to properly sense and respond to inflammatory cues constrains tissue expansion and predisposes to metabolic dysfunction [31,32,33].

Thus, WAT inflammation may represent a double-edged sword: maladaptive when chronic and excessive, yet potentially beneficial when engaged in controlled remodeling processes. Deciphering this paradox is essential, not only to resolve the causal interplay between inflammation and insulin resistance, but also to harness immunometabolic pathways as therapeutic targets in obesity and type 2 diabetes. This review explores the immune identities of adipocytes and preadipocytes, and their roles in regulating immune homeostasis within WAT under both physiological and pathological conditions.

## 2. Adipocyte Heterogeneity and Shifts in Cellular Identity During Obesity

Inflamed WAT in obesity includes aberrant lipid storage, impaired insulin responsiveness, and progressive tissue fibrosis. Considerable attention has been directed toward alterations in the stromal composition during overnutrition. Particularly, the emergence of fibro-inflammatory progenitor populations [4,34,35,36,37] and the infiltration of pro-inflammatory immune cells within the epididymal depot both reduce the proportion of adipocytes from ~30% in lean states to only about ~10% following high-fat diet (HFD) feeding [38] (Figure 1 and Figure 2). Yet despite decades of study, the true phenotype of mature adipocytes in obesity remains incompletely defined. This gap stems largely from the technical difficulty of purifying mature adipocytes, which are fragile, buoyant, and prone to rupture. Classic approaches, such as the Rodbell method, lack the precision required to isolate uncontaminated adipocyte populations [39]. To overcome the problem of cellular heterogeneity within intact tissues, several specialized isolation strategies have been developed. Laser-capture microdissection can harvest highly pure populations of rare cells [40], but it demands substantial expertise, costly instrumentation, and suffers from extremely low throughput. Fluorescence-activated cell sorting (FACS) offers another route when unique surface markers or Cre-dependent fluorescent reporters are available. However, the enzymatic dissociation required for FACS can itself distort cellular state [41]. To address these challenges, the team of Evan Rosen introduced NuTRAP (Nuclear tagging and Translating Ribosome Affinity Purification), a transgenic mouse system enabling the simultaneous isolation of cell-type–specific translating mRNA and chromatin from complex tissues [42]. Using NuTRAP, they define the transcriptional and epigenomic signatures of distinct adipocyte populations in vivo, revealing substantial divergence from whole-tissue profiles and from commonly used in vitro adipocyte cell lines [42]. Additionally, emerging technologies, including single-cell RNA sequencing and spatial transcriptomics, are now illuminating, with unprecedented precision, how adipocytes respond to the metabolic stresses of obesity [38,43,44,45,46,47].

Beyond compositional alterations, adipocytes undergo a profound transcriptional reprogramming toward a fibro-inflammatory identity (Figure 2). Comparative transcriptomics analyses between lean and obese mature adipocytes reveal a robust upregulation of pro-inflammatory cytokines, lysosomal program and stress-response genes, and ECM components, collectively driving chronic inflammation [38,42,44] (Figure 1 and Figure 2). This maladaptive remodeling, involving a shift in cellular identity, curtails adipogenic renewal and constrains healthy WAT expansion [33,35]. Such loss of identity carries systemic consequences. Decades of work, dating back to the 1970s, have demonstrated that larger adipocytes are intrinsically less insulin-responsive than their smaller counterparts [48,49]. They exhibit reduced insulin-stimulated glucose uptake (via GLUT4) and diminished suppression of fat breakdown (lipolysis) [48,49,50]. The resulting fatty-acid spillover promotes ectopic lipid deposition in the liver, skeletal muscle, kidneys, and vasculature, driving lipotoxicity, altered glucose metabolism, and systemic insulin resistance. As hypertrophic adipocytes also lose their capacity to store fatty acids efficiently, WAT becomes a net contributor to circulating lipids [51,52,53].

Spatial transcriptomics of human subcutaneous WAT has profoundly reshaped our understanding of adipocyte heterogeneity [43]. Only a restricted subset of adipocytes remains insulin-responsive, and the abundance of this subset correlates tightly with whole-body insulin sensitivity. Thus, metabolic health may depend on a privileged minority of functionally preserved adipocytes rather than the average state of the tissue. Spatial analyses revealed striking micro-organizations: pro-inflammatory adipocytes cluster into self-reinforcing neighborhoods that may serve as localized hubs of chronic inflammation [43,54]. Moreover, adipocyte size correlates with leptin (LEP) expression only within the Adipo*^LEP^* cluster, suggesting a potential size-dependent leptin feedback loop with central energy regulation [43]. Conversely, the Adipo*^SAA^* cluster expresses high levels of *RBP4* and *SAA1/2* [43], potent pro-inflammatory effectors, positioning this population as an inflammatory amplifier [55,56,57]. These findings support a view of obese WAT as a patchwork of spatially organized endocrine and inflammatory adipocyte niches, not a uniform depot of lipid-storing cells.

At the molecular core of this reprogramming lies the reduction in PPARγ activity. PPARγ is a nuclear hormone receptor and the master transcriptional regulator of adipocyte biology, controlling not only adipocyte differentiation but also the long-term survival and metabolic activity of mature adipocytes [58,59,60,61,62,63]. Under nutritional stress, its activity is markedly reduced in WAT [64,65,66,67]. Genome-wide binding analyses show that PPARγ occupancy is globally diminished in adipocytes from HFD-fed mice, with most sites downregulated [38]. These lost sites cluster at genes critical for adipokine secretion, insulin signaling, and lipid handling, precisely the pathways whose dysregulation defines the obese state. Whether this decline in binding reflects reduced PPARγ protein abundance, altered ligand availability, or cofactor dysregulation remains unresolved [38]. Mechanistically, PPARγ exerts potent anti-inflammatory effects beyond canonical transcriptional activation, directly repressing pro-inflammatory signaling by interfering with NF-κB, AP-1, and STAT pathways [68,69,70]. Collectively, these findings position the fibro-inflammatory transcriptional switch and collapse of PPARγ-driven networks as a pivotal nexus linking adipocyte dysfunction to the systemic pathogenesis of obesity and metabolic disease [38] (Figure 2).

## 3. What Drives the Inflammatory Program in Adipocytes in Obesity?

When caloric intake persistently exceeds energy expenditure, the brain–adipocyte feedback axis collapses under metabolic overload. Chronic hyperinsulinemia, hyperleptinemia, and alterations in catecholaminergic signaling drive persistent adipocyte expansion, pushing cells until their structural and metabolic limits [71,72,73,74]. Although the notion of a structural limit for adipocyte lipid storage has gained little mechanistic basis in experimental studies, it is generally accepted that beyond this threshold, anabolic force becomes deleterious, and the adipocyte shifts from storage to stress, from growth to inflammation [3]. Multiple stressors converge at this inflection point: mechanical strain, oxidative and ER stress, yet hypoxia emerges as the earliest and most potent trigger [3]. As adipocytes enlarge, oxygen diffusion lags behind cellular demand, creating micro zones of hypoxia within expanding WAT. The transcription factor HIF-1α, normally degraded via prolyl hydroxylase–mediated hydroxylation, stabilizes and activates a hypoxic gene program that drives angiogenesis, macrophage recruitment, oxidative stress, and fibrosis [75,76]. Genetic models confirm its causal role: adipocyte-specific HIF-1α knockout mice are protected from obesity-induced inflammation and systemic insulin resistance [77]. Importantly, hypoxia is not merely a downstream consequence of obesity; it arises early. Within days of overnutrition, adipocyte respiration becomes uncoupled, oxygen consumption surges, and local hypoxia develops, coinciding with early WAT inflammation [77,78]. Saturated fatty acids activate *adenine nucleotide translocase 2* (ANT2 encoded by the *SLC25A5* gene), a mitochondrial exchanger for ADP/ATP across cytoplasm and mitochondria, promoting proton leak and inefficient oxidative phosphorylation [77]. The paradox: more oxygen consumed, less ATP produced, generating a stress signal powerful enough to ignite inflammation. Simultaneously, excess fatty acid influx stimulates ANT2, the mitochondrial ADP/ATP carrier, escalating oxygen demand and fibrotic remodeling. Deletion of ANT2 reduces oxygen consumption while preserving mitochondrial integrity, shielding WAT from hypoxia-driven inflammation and fibrosis [79]. Collectively, these findings identify hypoxia as the molecular spark that transforms adipocytes from passive energy reservoirs into active instigators of inflammation, collapsing local oxygen homeostasis and destabilizing systemic metabolism (Figure 3).

This mitochondria-driven hypoxic state is accompanied by profound metabolic reprogramming in mature adipocytes. It is now well established that metabolism and gene expression are tightly interconnected: intermediary metabolites serve as substrates for chromatin-modifying enzymes and transcriptional regulators [80]. Among these metabolic regulators, glutamine has recently emerged as a pivotal immunometabolic node linking obesity to adipocyte inflammation. In adipocytes, glutamine depletion increases UDP-GlcNAc and enhances O-GlcNAcylation of chromatin-bound proteins near inflammatory loci, thereby reinforcing transcription of pro-inflammatory genes [81,82]. In obesity, glutamine availability declines while glutaminase (GLS) activity rises, accelerating glutamine catabolism to glutamate, a metabolic shift that amplifies inflammatory signaling [83]. Crucially, HIF-1α directly upregulates GLS, establishing a mechanistic bridge between hypoxia-induced mitochondrial stress and glutamine-driven inflammatory reprogramming [84]. Elevated GLS activity enhances glutamate synthesis and its export via the cystine/glutamate antiporter xCT (Slc7a11). This glutamate-rich microenvironment, in turn, activates hypoxia-induced CXCL12, which recruits natural killer (NK) cells. NK cells respond by producing interferon-γ (IFN-γ) via mGluR5 activation triggered by extracellular glutamate. IFN-γ then reinforces xCT and CXCL12 expression in adipocytes, creating a self-perpetuating adipocyte-NK cell feedback loop that sustains macrophage activation and metabolic dysfunction [84] (Figure 4).

Hypoxic adipocytes also undergo a glycolytic shift, increasing lactate production to sustain ATP generation. Remarkably, adipocytes release substantial amounts of lactate even under normoxic conditions [85,86]. This “Warburg-like” metabolic adaptation persists in insulin-resistant adipocytes, suggesting that lactate production is a core feature of adipocyte metabolism rather than a byproduct of glucose overflow [85,86]. Functionally, lactate acts as a signaling metabolite. Adipocyte-derived lactate stabilizes HIF-1α in macrophages by directly competing with α-ketoglutarate for binding to prolyl hydroxylase domain-containing 2 (PHD2), thereby promoting IL-1β expression. Accordingly, *Ldha* deletion, which blocks pyruvate-to-lactate conversion, confers protection against obesity-induced insulin resistance and inflammation, concomitantly reducing macrophages and IL-1β production in WAT [87]. Thus, lactate serves as a paracrine amplifier of inflammation linking adipocyte metabolism to immune cell activation (Figure 4).

Other studies have uncovered additional bioenergetic nodes in this inflammatory cascade. Phosphocreatine metabolism, traditionally regarded as a cellular energy buffer, emerges as a key determinant of adipocyte function. Creatine kinase B (CKB) catalyzes the reversible conversion of creatine to phosphocreatine, thereby maintaining the ATP/ADP ratio. In obesity, phosphocreatine turnover is disturbed, and CKB deficiency elevates ATP/ADP ratios and provokes adipose inflammation, identifying CKB as a gatekeeper of metabolic homeostasis [88]. Mechanistically, ER stress represses *CKB* transcription through the XBP1s–DNMT3A axis, promoting promoter methylation and reduced *CKB* gene expression [89]. Perturbation of this pathway upregulates pro-inflammatory cytokines such as CCL2 (MCP1), thereby integrating ER stress, creatine metabolism, and inflammatory signaling in white adipocytes [88,89,90] (Figure 4).

Downstream of metabolic reprogramming, the hypoxic adipocyte actively reshapes its microenvironment. HIF-1α activation in hypertrophic adipocytes drives excessive production of ECM components, particularly collagens I, IV, and VI, contributing to fibrosis and tissue stiffening [34,91,92]. What initially serves as an adaptive attempt to reinforce tissue integrity quickly degenerates into pathology: ECM accumulation is believed to restrict adipocyte expansion, perpetuating a vicious cycle of mechanical compression, local hypoxia, and chronic inflammation [93]. Functional studies reinforce this causal relationship. Collagen VI deletion in mice markedly reduces adipose inflammation and enhances insulin sensitivity despite persistent hypertrophy [94], whereas its excessive pericellular deposition fosters macrophage infiltration and crown-like structure (CLS) formation. Similarly, in humans, collagen VI upregulation correlates tightly with inflammation, insulin resistance, and metabolic dysregulation [95]. As fibrosis stiffens the adipose matrix, mechanical constraint itself becomes an inflammatory cue. Compression of hypertrophic adipocytes distorts mechano-transduction, disrupts insulin signaling, stimulates adipokine secretion, and perturbs lipid handling [23,96]. Remarkably, ex vivo compression alone recapitulates the inflammatory phenotype, demonstrating that mechanical load is sufficient to convert adipocytes into pro-inflammatory effectors [96] (Figure 4).

Together, these insights redefine adipocytes as active architects of their own inflammatory niche. By coupling hypoxic signaling to ECM remodeling and mechano-transduction, the adipocyte transforms from a passive energy reservoir into a dynamic inflammatory hub, orchestrating the metabolic, mitochondrial, and mechanical stress responses that drive systemic disease.

## 4. Adipokines: Direct Modulators of the Immune Status Within White Adipose Tissue

Since its initial identification in the 1980s by the Spiegelman lab, Adipsin emerged as the first characterized adipokine [97,98]. Adipokines, proteins secreted by adipocytes into the bloodstream, play key roles in regulating metabolic and immune functions. Subsequent studies revealed that Adipsin is identical to Complement Factor D (CFD), a crucial component in the alternative pathway of the complement system [99]. While most complement components are synthesized by hepatocytes, macrophages, or endothelial cells, Adipsin is almost exclusively produced by WAT under the control of PPARγ activation [60,100]. In experimental models, Adipsin-deficient mice (*Adipsin*^−/−^) fed a high-fat diet gained less weight than wild-type counterparts, exhibiting reduced inflammation in WAT, fewer macrophage CLS, and diminished mast cell numbers [101]. However, despite lower adiposity, these mice developed impaired glucose tolerance after 16 weeks of HFD feeding without changes in insulin sensitivity [101]. The cause of this glucose intolerance was linked to defective pancreatic β-cell function and reduced insulin secretion. Strikingly, restoring Adipsin levels in *db*/*db* mice using a recombinant adenoviral vector improved fasting blood glucose levels and glucose tolerance, an effect likely mediated by the protein C3a. This body of research underscores the complex role of Adipsin in both inflammation and glucose homeostasis. Although its complete in vivo function remains unclear, recent findings suggest that Adipsin also supports insulin secretion by pancreatic β-cells and protects these cells from apoptosis [101,102].

In 1993, TNFα was identified as a key pro-inflammatory cytokine produced by WAT, particularly in the context of obesity [103]. Early studies suggested adipocytes contributed to its production, but it is now clear that adipocytes are not the major source of TNFα. Instead, the predominant producers are macrophages and other immune cells within the stromal vascular fraction (SVF), which markedly expand in obese WAT. These immune-cell–derived TNFα signals act on neighboring adipocytes to amplify inflammation, impair insulin signaling, and reinforce metabolic dysfunction [104,105,106]. TNFα plays a critical role in impairing insulin signaling by suppressing key components such as the insulin receptor, insulin receptor substrate 1 (IRS-1), and GLUT4, thereby reducing insulin-stimulated glucose uptake in adipocytes and other insulin-sensitive tissues [107,108,109]. Additionally, in WAT, TNFα promotes the expression of genes involved in endoplasmic reticulum and oxidative stress responses, contributing to mitochondrial dysfunction and chronic inflammation [110]. Notably, TNFα signaling through TNF receptor 1 (TNFR1) can trigger pro-apoptotic pathways, leading to caspase-8 and caspase-3 activation, adipocyte cell death, and further exacerbation of adipose tissue inflammation [111]. Studies in animal models have demonstrated improved insulin sensitivity following TNFα neutralization, underscoring its pivotal role in metabolic dysfunction associated with obesity [103,112,113]. A similar upregulation of TNFα in WAT has been observed in humans with obesity, suggesting a direct link between higher TNFα levels, increased adiposity, and insulin resistance [113]. However, clinical trials using TNFα blocking antibodies in insulin-resistant humans have been disappointing, as single injections failed to elicit improvements in metabolic and clinical markers of insulin resistance [114,115,116].

One of the most significant advancements in understanding the role of WAT in regulating immunometabolism was the identification of the genetic forms of obesity and related syndromes [117]. In 1994, Friedman and collaborators discovered the *ob* gene, which encodes leptin, a cytokine-like hormone named from the Greek word “leptos,” which means “thin” [118]. Leptin is “exclusively” secreted by adipocytes [118]. Mutations in the *ob* gene result in a lack of leptin production, leading to excessive eating, weight gain, and disruptions in fertility and thermoregulation [118,119,120]. Exogenous Leptin administration can correct these symptoms in *ob*/*ob* mice [120,121]. Likewise, in *ob*/*ob* mice, leptin treatment also reduces food intake and causes significant weight loss in humans with leptin gene mutations [121,122,123,124,125]. However, individuals with common forms of obesity typically exhibit elevated plasma leptin levels compared to lean controls, reflecting a state of leptin resistance [126,127]. Recent studies by Scherer and colleagues revealed that elevated leptin levels can contribute to metabolic disorders [128]. Conversely, partial reduction in circulating leptin in obesity can restore leptin sensitivity in the hypothalamus, leading to reduced weight gain and enhanced insulin sensitivity [128]. Beyond its metabolic functions, leptin exerts a broad immunomodulatory effect [129]. Leptin deficiency increases susceptibility to severe responses to lipopolysaccharide (LPS) or TNFα, although this effect can be partially alleviated by leptin treatment [130]. In leptin-deficient mice, macrophages exhibit impaired phagocytosis and altered cytokine production [131,132]. Leptin treatment of CD4+ T cells enhances the production of pro-inflammatory cytokines originating from T helper 1 (Th1), such as interferon gamma (IFN-γ) and IL-2, and suppresses production of the T helper 2 (Th2) cytokine, IL-4 [133]. Additionally, activated CD4+ T cells from T cell-specific leptin receptor knockout mice produce markedly less IFN-γ compared to wild-type counterparts [134]. These findings collectively suggest that leptin drives pro-inflammatory cytokine production in CD4+ T cells. Furthermore, leptin facilitates the differentiation of pro-inflammatory CD4+ Th1 cells. These data reinforce the dual metabolic and immune roles of leptin (Figure 5).

In contrast to leptin, adiponectin is an essential adipokine that is markedly reduced in obesity and exhibits anti-inflammatory effects and insulin-sensitizing effects [135,136,137,138,139]. Produced almost exclusively by mature adipocytes, adiponectin plays a central role in modulating inflammation and metabolism [139]. In *ob*/*ob* mice that overexpress adiponectin, there is an enhanced capacity for subcutaneous WAT expansion, which is associated with attenuated inflammation and reduced ectopic lipid accumulation in the liver and skeletal muscle [140]. These metabolic improvements lead to improved insulin sensitivity despite the increased WAT mass [140]. Within the innate immune system, adiponectin primarily affects macrophages by promoting anti-inflammatory M2-like polarization and decreasing the abundance of pro-inflammatory M1-like macrophages [141]. Additionally, adiponectin reduces neutrophil phagocytic capacity, promotes neutrophil survival, and limits the production of interferon γ (IFNγ) and interleukin 17 (IL-17) in CD4+ T cells [142]. These effects highlight adiponectin’s role in counterbalancing inflammation and improving metabolic outcomes (Figure 5).

Adipocytes recruit immune cells into WAT by producing a spectrum of pro-inflammatory cytokines and chemokines such as interleukin (IL)-6, IL8, CXCL2, and chemokine (C-C motif) ligand 2 (CCL2)/monocyte chemoattractant protein 1 (MCP1), whose gene expression and protein secretion are markedly increased in obesity [3]. In both humans and mice, adipocyte-derived MCP1 is markedly increased and represents a key mediator of macrophage infiltration into AT during obesity [143,144,145]. MCP1 appears to be particularly important, as it has been proposed to initiate adipose inflammation by recruiting inflammatory cells from the bloodstream into WAT [144,146]. Studies in mice demonstrate that MCP1 production and signaling are determinants for the onset and progression of WAT inflammation [20]. Although a number of different cell types in WAT may produce MCP1, adipocytes are of particular interest, as adipocyte-derived MCP1 can sustain local inflammation independently of macrophages or leukocytes in human WAT [147].

## 5. Role of Adipocyte-Derived Extracellular Vesicles in White Adipose Tissue Inflammation

Adipocytes also secrete EVs, which serve as a novel and highly dynamic mode of intercellular communication with other cells and tissues [148,149,150,151]. These EVs contain a variety of bioactive substances, including adipokines, lipids, microRNAs (miRNAs), and even mitochondria [152,153,154]. These components play crucial roles in influencing inflammation and metabolism both locally and throughout the body. One of the most abundant adipokines found in WAT-derived EVs is adiponectin, which retains its insulin-sensitizing and anti-inflammatory properties when delivered through EVs [152].

EVs are divided into two major subtypes: exosomes, which are 40–150 nm in size and formed through endosomal trafficking and exocytosis, and microvesicles, which are 100–1000 nm and generated by direct budding from the cell membrane [155]. EV release is enhanced in obesity but suppressed by caloric restriction or lipodystrophy [150]. A HFD, particularly enriched with palmitate, strongly stimulates EV release [150]. EVs are taken up by target cells via endocytosis, pinocytosis, or phagocytosis, guided by surface adhesion molecules that confer cell-type specificity [156]. Adipocytes represent a major source of circulating miRNAs, with over 60% of EV-derived miRNAs in mice originating from adipocytes [157]. This allows adipocytes to control protein production in other cells through post-transcriptional regulation.

In mice, miRNAs within adipocyte-derived EVs regulate the inflammasome activation, IL-1β production, and macrophage polarization, and influence insulin secretion from pancreatic β cells, thereby linking adipocyte-derived EV signaling to systemic metabolic and inflammatory homeostasis [158,159]. In humans, this regulatory axis appears to shift toward pathology in the setting of obesity. EVs released from adipose tissue, particularly from visceral depots, of individuals with obesity amplify inflammatory responses by stimulating macrophages to produce pro-inflammatory cytokines [160]. These findings underscore the dual nature of adipocyte-derived EVs, which can act as finely tuned regulators of metabolic homeostasis under physiological conditions but become powerful drivers of chronic inflammation and metabolic disease when adipose tissue function is disrupted.

A particularly striking discovery is that adipose-derived EVs can carry mitochondria. This finding extends the concept of mitochondria from static intracellular energy producers to transferable organelles capable of intercellular exchange—a phenomenon termed intercellular mitochondrial transfer [161]. Mitochondrial extrusion via EVs, demonstrable in organelle-tracking mouse models, depends strongly on mitophagy and other quality-control pathways. It is typically activated under conditions of autophagy deficiency, serving as an alternative mechanism to remove dysfunctional organelles. Enlarged adipocytes with mitochondrial dysfunction and impaired autophagic flux [162] are major contributors to mitochondria-containing EVs, which are enriched in phosphatidylglycerol, a mitochondrial phospholipid precursor for cardiolipin synthesis [163]. Importantly, mitochondria-bearing EVs are actively internalized by resident macrophages, promoting their anti-inflammatory polarization [164]. This regulatory axis, however, becomes disrupted under nutrient excess: dietary lipid overload increases the efflux of adipose-derived EVs into the circulation [165], leading to phosphatidylglycerol dysregulation and contributing to the systemic inflammatory milieu characteristic of obesity [166].

## 6. Adipocyte Antigen Presentation and Adaptive Immunity

Both adipocytes and macrophages play crucial roles in shaping adaptive immune responses within WAT. Selective knockout of the MHCII pathway, responsible for antigen presentation, has revealed that adipocytes and macrophages contribute comparably to this immune function [167,168]. Interestingly, in both humans and mice, the expression of the MHCII pathway in adipocytes rises sharply within as little as two weeks of a HFD in lean subjects [169]. This early activation occurs in both subcutaneous and visceral fat depots, indicating that adipocytes act as early initiators of immune activation and inflammatory signaling in response to dietary stress [169]. The importance of adipocyte antigen presentation has been further underscored by genetic studies in murine models. Adipocyte-specific deletion of H2Ab1, a key component of the MHCII antigen presentation, significantly attenuates inflammation in WAT and improves insulin sensitivity under obese conditions [170]. Remarkably, this phenotype was independent of the T regulatory cells (Treg) abundance within WAT, suggesting that adipocyte-derived MHCII signaling specifically contributes to pro-inflammatory changes associated with obesity [170]. In contrast to macrophages, adipocytes appear to be the primary responders initiating this process, whereas macrophage participations emerge at later stages of pathological progression.

## 7. Adipocyte Apoptosis as a Catalyst of Inflammation in White Adipose Tissue

Adipocyte death is a critical early event that triggers macrophage infiltration into WAT, thereby contributing to the onset and progression of insulin resistance in obesity, as observed in both humans and mice [171]. Dying adipocytes release a spectrum of pro-inflammatory signals that recruit immune cells to the tissue, driving local inflammation [172]. In the obese state, this inflammatory milieu sustains and amplifies adipocyte demise by engaging multiple programmed cell death pathways, including pyroptosis, apoptosis, and necroptosis, collectively referred to as PANoptosis, or inflammatory cell death [173,174,175,176,177]. This phenomenon is particularly prominent in obesity, where macrophages surround apoptotic adipocytes, forming characteristic CLS [178]. Among the key mediators in this process is the NOD-like receptor (NLR), a family of pattern recognition receptors (PRRs) expressed in macrophages, which sense damage-associated molecular patterns (DAMPs) released from stressed or dying adipocytes [179,180]. In macrophages, NLR activation leads to the formation of the inflammasome, a multiprotein complex that activates caspase-1, leading to the cleavage of pro-IL-1β and pro-IL-18 [181]. Diet-induced obesity enhances the production of caspase-1 and IL-1β in WAT, thereby exacerbating local inflammation [182].

One of the key molecular regulators of adipocyte apoptosis is caspase-8, which has both apoptotic and non-apoptotic functions. Caspase-8 is a key initiator of the death receptor (DR)-mediated extrinsic apoptosis pathway [183]. In this signaling cascade, Fas, a prototypical DR, binds to its ligand FasL, leading to the recruitment of Fas-associated death domain (FADD) and caspase-8, forming the death-inducing signaling complex (DISC). Caspase-8 is subsequently activated through auto-cleavage and, in turn, activates caspase-3, which executes programmed cell death [184]. Experimental evidences indicate that caspase-8 expression is increased in the WAT of both humans and mice in obesity, where it correlates with insulin resistance [185]. The FAT-ATTAC (Fat Apoptosis Through Targeted Activation of Caspase-8) mouse model has been instrumental in elucidating the role of caspase-8-mediated apoptosis in WAT. In this model, forced expression and dimerization-induced activation of caspase-8 trigger adipocyte death, leading to local inflammation and impaired glucose tolerance [186]. As a result of caspase-8 activation, there is an increase in macrophage infiltration into the WAT [187]. In contrast, adipocyte-specific knockdown of caspase-8 confers protection against glucose intolerance and weight gain in mice fed HFD. These mice display reduced WAT inflammation and diminished activation of both canonical and non-canonical NF-κB signaling pathways [185]. This indicates that caspase-8 plays a critical role in regulating both apoptosis and inflammation within WAT. Further research underscores the role of FADD, the adaptor protein involved in death receptor signaling, in modulating WAT inflammation and metabolism. Adipocyte-specific deletion of FADD reduces inflammation and enhances fatty acid oxidation via PPAR-α activation, thereby conferring resistance to HFD-induced obesity [188]. Moreover, FADD haploinsufficient mice (*Fadd*^+/−^) exhibit reduced WAT mass and downregulated expression of adipogenic and lipogenic genes [189]. Cellular studies corroborate these findings; in vitro inhibition of FADD impairs adipocyte differentiation and suppresses the expression of adipogenic and lipogenic genes in cultured adipocytes. Conversely, overexpression of FADD accelerates adipocyte apoptosis [189]. These findings suggest that FADD not only promotes apoptosis but also regulates lipid metabolism and adipocyte differentiation. In conclusion, adipocyte death, driven by molecular regulators such as caspase-8 and FADD, fuels local inflammation and impacts systemic metabolism.

## 8. Preadipocytes: Immune-like Cells

In murine WAT, two main functionally distinct groups of adipose progenitor cells (APCs) residing within major WAT depots (inguinal and perigonadal) have been identified. The first group, commonly referred to as "preadipocytes," exhibits a robust potential for adipogenesis both in vitro and in vivo, and is characterized by enriched expression of *Pparg* and other markers associated with terminal adipocyte differentiation [33,190,191,192]. The second group, known as "fibro-inflammatory" progenitors (FIPs), shows less commitment to adipocyte differentiation [36,37,193]. Instead, these cells are distinguished by a fibro-inflammatory phenotype, actively secreting extracellular matrix components and pro-inflammatory cytokines [36]. These APCs are predominantly found in interstitial niches, often in proximity to blood vessels and lymphatics [190,193]. This strategic spatial arrangement enables them to sense metabolic cues and interact with vasculature-associated immune cells.

The signaling mechanisms from adipose stromal cells, including APCs, are critical in maintaining niches that support type 2 immune cells. Type 2 immunity mediates defense against helminths, contributes to allergic inflammation, and supports tissue homeostasis and repair. It involves innate cells such as group 2 innate lymphoid cells (ILC2s) and eosinophils, as well as adaptive type 2 helper T (Th2) cells, all characterized by the production of interleukins IL-4, IL-5, and IL-13 [194]. For instance, eosinophil recruitment into WAT is modulated by the chemokine CCL11, which is secreted by APCs in response to IL-4 and IL-13 [195] or by mature adipocytes following sympathetic activation [196]. IL-33 has emerged as a pivotal mediator in these niches, particularly in regulating anti-inflammatory Tregs and ILC2s [197]. Within WAT, PDGFRα+Pdpn+ APCs have been identified as the primary source of IL-33 [198,199]. These APCs not only facilitate the accumulation of anti-inflammatory lymphocytes but also actively sustain a TH2 immune milieu, crucial for WAT homeostasis [198]. The switch in IL-33 source in aged WAT is associated with a senescence-like phenotype of ILC2, indicating that the cellular context of IL-33 production might impact ILC2 functionality and contribute to age-related metabolic dysfunction [200]. Thus, APCs serve as key orchestrators of type 2 immunity, playing an indispensable role in preserving WAT integrity and immune balance.

On the other hand, type 2 immune reaction is vital for WAT remodeling and plasticity of preadipocytes [194]. For example, IL-4Rα signaling in PDGFRα+ cells promotes the expansion of adipocyte precursors [201]. Additionally, adipose eosinophils, macrophages, and Th2 cells produce TGFβ cytokines [202,203,204], which drive proliferation over differentiation in APCs. However, once APCs commit to the adipocyte lineage (ICAM1+ preadipocytes), they become resistant to TGFβ’s proliferative effects [205]. In addition, TGFβ3 has been identified as a specific regulator that enhances adipocyte precursor proliferation, thereby contributing to the homeostatic control of adipocyte number in vivo [206].

Thus, type 2 cytokines dynamically regulate the balance between APC proliferation and differentiation, enabling the tissue to adapt rapidly to changing conditions. This includes shifting toward hyperplasia rather than hypertrophy in response to excess calories. This adaptive mechanism highlights the critical role of type 2 inflammation in preserving the structural and functional flexibility of WAT.

## 9. Conclusions and Perspectives

White adipose tissue is now recognized as a dynamic immunometabolic organ in which adipocytes and their progenitors actively shape both local and systemic immune responses. Far from passive lipid stores, adipocytes act as endocrine and paracrine hubs, releasing adipokines, cytokines, lipids, and extracellular vesicles that regulate inflammation, insulin sensitivity, and tissue homeostasis. In obesity, chronic metabolic stress, hypoxia, and adipocyte death reprogram these cells toward a pro-inflammatory state, promoting immune cell recruitment and amplifying local and systemic metabolic dysfunction.

Despite these advances, key questions remain unresolved. What signals initiate adipose inflammation? Which cell types respond first, and how is this response coordinated over time? How do adipocytes, APCs, immune cells, and stromal elements communicate to support adipose expansion, and when does this initially adaptive response become maladaptive and pathological? Addressing these questions requires reframing adipocytes as active drivers, rather than passive targets, of immune–metabolic crosstalk. While this adipocyte-centric perspective simplifies an inherently complex system, it offers a powerful framework for understanding WAT dysfunction in obesity. Ultimately, resolving these interactions will require integrative, high-resolution approaches capable of capturing the spatial and temporal diversity of adipose-resident cells. Such insights are essential to understand how tissue homeostasis and plasticity are maintained and how chronic metabolic stress and aging disrupt this balance, driving fibrosis, immune dysregulation, and loss of metabolic resilience. Elucidating when and how this transition occurs will be critical for developing strategies to restore adipose tissue health in metabolic disease.

## Figures and Tables

**Figure 1 biomolecules-16-00059-f001:**
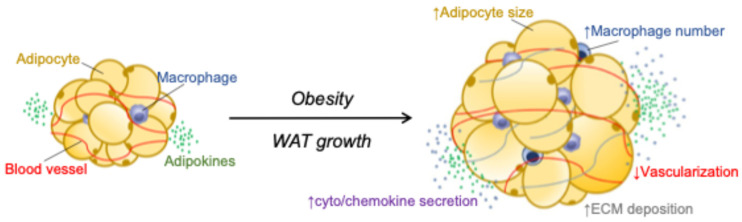
Remodeling of the adipose niche in obesity. With increasing body mass index (BMI), white adipose tissue (WAT) undergoes extensive remodeling. In lean states, WAT maintains a balanced immune landscape composed predominantly of resident macrophages, eosinophils, and regulatory T cells (Tregs), supporting healthy adipocyte function and tissue homeostasis. Chronic overnutrition disrupts this equilibrium: adipocytes enlarge, adipokine secretion becomes aberrant, and the stromal compartment is overtaken by pro-inflammatory immune cells, including pro-inflammatory macrophages, mast cells, and diverse effector T-cell subsets. In parallel, excessive extracellular matrix deposition leads to progressive fibrosis, which stiffens the tissue, restricts adipocyte expandability, and amplifies inflammation. Together, these alterations transform WAT into a dysfunctional, highly inflammatory, and fibrotic environment characteristic of obesity.

**Figure 2 biomolecules-16-00059-f002:**
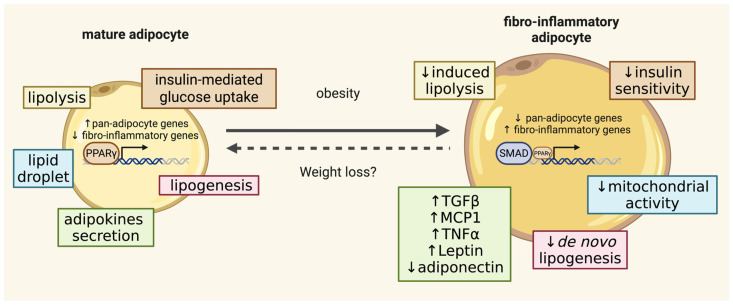
Obesity profoundly alters adipocyte biology. Enlarged adipocytes undergo profound functional decline. They display reduced sensitivity to hormonal and nutrient cues, impaired insulin-stimulated glucose uptake, decreased de novo lipogenesis, blunted hormonally induced lipolysis, and markedly diminished adiponectin secretion. In parallel, hypertrophic adipocytes secrete elevated levels of cytokines and chemokines, fostering immune-cell recruitment and establishing a chronic inflammatory milieu within the tissue. These cellular defects are compounded by substantial remodeling of the extracellular matrix (ECM). In obese WAT, the ECM becomes abnormally stiff, a defining hallmark of adipose pathology. Elevated stiffness correlates strongly with insulin resistance, impaired glucose metabolism, and heightened inflammation, likely because it restricts the adaptive remodeling required for healthy tissue expansion. Created in BioRender. Lec, S. (2026) https://BioRender.com/kthl9nn. (accessed on 2 January 2025).

**Figure 3 biomolecules-16-00059-f003:**
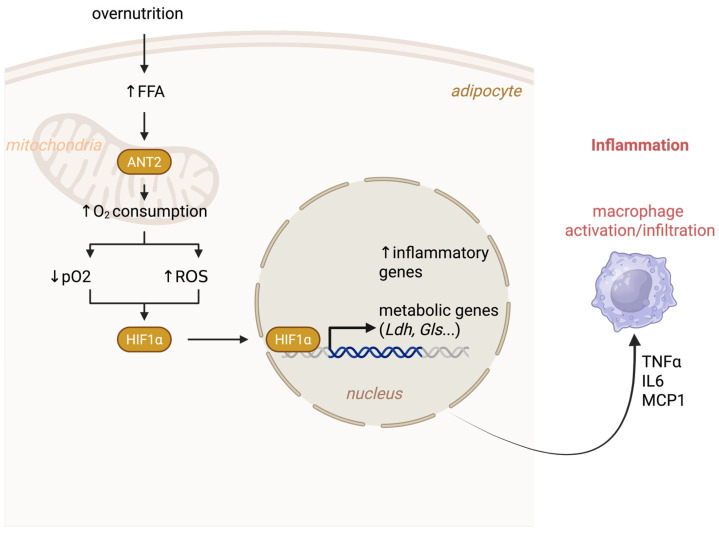
Mitochondrial Uncoupling and Hypoxia as Core Drivers of Adipocyte Dysfunction in Obesity. In lean adipose tissue, adipocytes are well oxygenated by the surrounding vasculature, allowing efficient coupled mitochondrial respiration. Protons are pumped from the mitochondrial matrix into the intermembrane space, generating the membrane potential required for ATP synthesis. Under these conditions, the adenine nucleotide translocase ANT2 remains minimally active, limiting proton leak and maintaining tightly coupled respiration—a metabolic state that preserves normal insulin sensitivity. In obesity, however, ANT2 activity increases, driving proton leak back into the mitochondrial matrix. This shift enhances uncoupled respiration, reduces mitochondrial membrane potential, and elevates oxygen demand despite a progressively inadequate oxygen supply. The resulting local hypoxia activates HIF1α, which in turn induces fibro-inflammatory gene programs and promotes metabolic reprogramming. Although initially adaptive, chronic activation of this pathway compromises adipocyte plasticity and accelerates tissue dysfunction. The persistent mismatch between oxygen availability and oxygen consumption triggers mitochondrial stress and fosters a fibro-inflammatory environment, ultimately contributing to insulin resistance, a defining feature of metabolic dysfunction in obese adipose tissue. FFA, free fatty acids; pO_2_, partial pressure of oxygen; HIF-1α, hypoxia-inducible factor 1 alpha; ROS, reactive oxygen species; ANT2, adenine nucleotide translocase 2; LDH, lactate dehydrogenase; GLS, glutaminase; TNFα, tumor necrosis factor alpha; IL-6, interleukin 6; MCP-1, monocyte chemoattractant protein 1 (CCL2). Created in BioRender. Lec, S. (2026) https://BioRender.com/mzxbcnh. (accessed on 2 January 2025).

**Figure 4 biomolecules-16-00059-f004:**
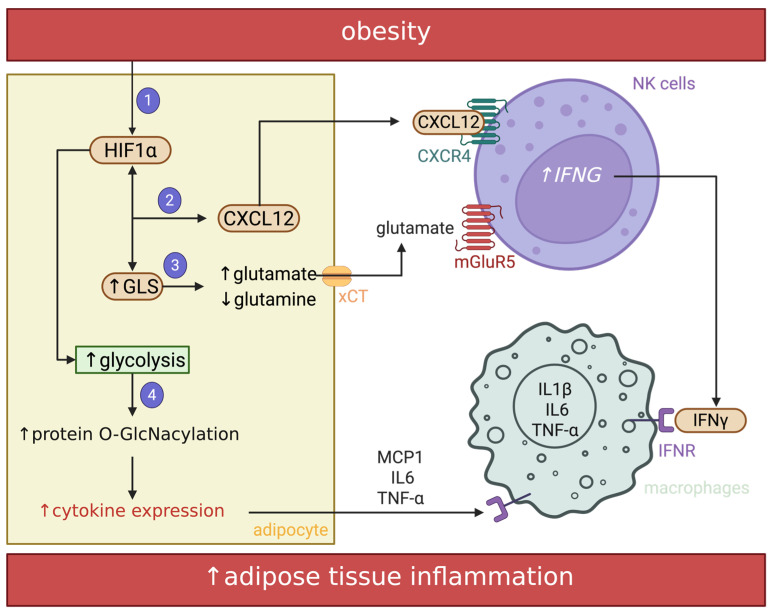
Metabolic circuit driving adipocyte inflammation. Nutrient overload induces local hypoxia in adipose tissue, leading to activation of HIF-1α (**1**) and the acquisition of a fibro-inflammatory adipocyte phenotype. This state is characterized by the overexpression of inflammatory genes, including CXCL12, which is secreted by adipocytes and binds to its receptor CXCR4 expressed on NK cells, thereby promoting IFN-γ production (**2**). Concomitantly, adipocytes undergo metabolic reprogramming marked by increased expression of glutaminase (GLS), enhancing glutamine hydrolysis to glutamate. Glutamate is then exported via the xCT transporter and activates NK cells through mGluR5, further stimulating IFN-γ production (**3**). IFN-γ subsequently promotes macrophage activation and the secretion of pro-inflammatory mediators. In parallel, glutamine depletion in adipocytes, together with increased glycolysis, leads to elevated protein O-GlcNAcylation, which enhances inflammatory gene expression. As a result, adipocytes secrete pro-inflammatory cytokines that further activate immune cells (**4**). Together, these pathways illustrate that adipocytes are not merely passive targets but are key active drivers of inflammation within adipose tissue. HIF-1α, hypoxia-inducible factor 1 alpha; CXCL12, C–X–C motif chemokine ligand 12; CXCR4, C–X–C chemokine receptor 4; NK cells, natural killer cells; IFN-γ, interferon gamma; GLS, glutaminase; xCT, cystine/glutamate antiporter (SLC7A11); mGluR5, metabotropic glutamate receptor 5; O-GlcNAcylation, O-linked β-N-acetylglucosamine modification; IL-6, interleukin 6; TNFα, tumor necrosis factor alpha; IL-1β, interleukin 1 beta. Created in BioRender. Lec, S. (2026) https://BioRender.com/mclp3ar. (accessed on 2 January 2025).

**Figure 5 biomolecules-16-00059-f005:**
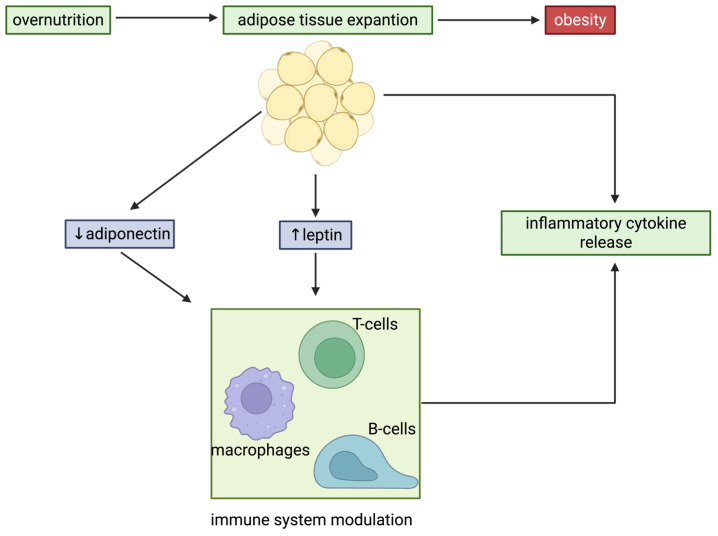
Leptin-driven immune inflammation during adipose tissue expansion. Overnutrition promotes adipose tissue expansion, leading to increased leptin expression and secretion by adipocytes. Adipocyte-derived leptin stimulates pro-inflammatory cytokine production by both innate and adaptive immune cells, thereby creating an inflammatory milieu that contributes to systemic metabolic dysregulation. In parallel, adipocyte secretion of adiponectin, an adipokine with anti-inflammatory properties, is reduced. Created in BioRender. Lec, S. (2026) https://BioRender.com/88e0jzr. (accessed on 2 January 2025).

## Data Availability

No new data were created or analyzed in this study.

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
