# Peer review of "How Adipocytes Orchestrate Inflammation Within Adipose Tissue?"

_biomolecules, 2025, doi:10.3390/biom16010059_

Round 1
Reviewer 1 Report
Comments and Suggestions for Authors
In the present Review, Higos and colleagues summarize how adipose tissue functions as a dynamic metabolic and endocrine organ that responds to nutrient excess by initiating a fibro-inflammatory adaptation that becomes maladaptive over time. This chronic response leads adipocytes to lose metabolic flexibility, adopt a fibro-inflammatory phenotype, and contribute to inflammation, insulin resistance, and metabolic dysfunction in obesity. This is a timely, well-written and interesting review showing an original adipocyte-centric framework that integrates diverse cellular and molecular cues to explain how fibro-inflammation reshapes adipose tissue homeostasis. Nonetheless, some specific points require to be amended.
Specific comments:
- Introduction, lines 70-73: It would be interesting to mention the concept of healthy immunometabolism that was firstly proposed by Wernstedt Asterholm and colleagues (Wernstedt Asterholm I et al, Cell Metab 2014, PMID: 24930973). This “healthy inflammation” scavenges the debris of dead adipocytes, enables adipogenesis as well as an appropriate extracellular matrix remodeling during the adipose tissue expansion.
- Section 4. Adipokines, lines 328-334: It is worth to mention that upon binding its receptor TNFR1 in human visceral adipocytes, TNF-α triggers proapoptotic signalling, resulting in caspase-8 and caspase-3 cleavage and activation, leading to adipocyte cell death and further contributing to adipose tissue inflammation in this fat depot (Rodríguez A et al, Diabetologia 2012, PMID: 22869322).
- Section 5. Role of adipocyte-derived extracellular vesicles, lines 407-410: In humans, adipose tissue-derived extracellular vesicles (EVs) from people living with obesity, particularly those originating from visceral adipose tissue, have been shown to exacerbate inflammation and promote the release of pro-inflammatory cytokines following interaction with macrophages (Lago-Baameiro N et al, J Transl Med 2025, PMID: 39979938).
- Section 7. Adipocyte Apoptosis: The chronic inflammatory milieu characteristic of adipose tissue in obesity can concurrently activate three programmed cell death pathways in adipocytes—pyroptosis, apoptosis, and necroptosis—collectively referred to as PANoptosis or inflammatory cell death (Giordano A et al, J Lipid Res 2013, PMID: 23836106; Deepa SS et al, Aging Cell 2018, PMID: 29696779; Rodríguez A et al, Cell Mol Immunol 2021, PMID: 34465884; Leven AS et al, Adipocyte 2021, PMID: 34743657).
Author Response
Comment 1: We sincerely thank the reviewer for this insightful suggestion. We have now incorporated the concept of healthy immunometabolism into the Introduction, emphasizing its role in orchestrating debris clearance, adipogenesis, and appropriate extracellular matrix remodeling during adipose tissue expansion, as originally described by Wernstedt Asterholm et al. This addition is highlighted in the revised manuscript.
Comment 2: We thank the reviewer for this important clarification. This mechanistic aspect of TNF-α signaling in human visceral adipocytes has now been added to Section 4, including its role in caspase-8– and caspase-3–mediated apoptosis and the consequent amplification of adipose tissue inflammation. The new text is highlighted in the revised manuscript.
Comment 3: We sincerely thank the reviewer for pointing out this recent and highly relevant study. We have now incorporated these findings into Section 5, highlighting the pathogenic potential of visceral adipose tissue–derived EVs in promoting macrophage-driven inflammation in humans with obesity. This addition is clearly indicated in the corrected manuscript.
Comment 4: We thank the reviewer for this valuable suggestion. We have now expanded Section 7 to explicitly describe PANoptosis as a central feature of adipocyte death in obesity, supported by the cited literature. This concept is now clearly articulated and highlighted in the revised manuscript.
Reviewer 2 Report
Comments and Suggestions for Authors
The review written by Higos, R. et al. provides a comprehensive summary of cellular and molecular cues that trigger fibro-inflammation and offers an overview of recent advances in this field. In general, the review is well-written and has provided new understanding on this topic. However, there are a few concerns that may provide opportunities for improvement.
- Please add citations of figures to the text.
- Figure 4 legend does not seem to match well with the diagram. Please revise the legend to better align the diagram. Also, it might be helpful to add abbreviations at the end of legend.
- Figures are concentrated in early sections. Perhaps add a few diagrams for later sections.
- Long sections (e.g., 2 and 3) may need to be broken down into a few subsections with titles to help readers’ understanding, such as 2.1., 2.2., 2.3. ….
- Citations are needed for line 315-316.
- Please explain what type 2 immune cells (line 500) are for readers who are not specialized in immunology.
- The last sentence says that future perspective should focus on … new pharmacological targets to prevent or treat type 2 diabetes. The reviewer found it disconnected as the whole review paper is on obesity and inflammation, no clear extension of obesity to type 2 diabetes. Please establish a clear connection between obesity and inflammation to type 2 diabetes.
Author Response
Comment 1: We thank the reviewer. Citations to the corresponding figures have now been added throughout the text where appropriate.
Comment 2: We thank the reviewer for pointing this out. Figure 4 has been revised, and the legend has been rewritten to more accurately reflect the content of the diagram. A list of abbreviations has also been added at the end of the legend for clarity.
Comment 3: We agree with this observation. To improve visual balance and support later sections of the manuscript, we have added a new Figure 5 illustrating the immunomodulatory role of leptin.
Comment 4: We thank the reviewer for this helpful suggestion. While we appreciate the proposed restructuring, we have chosen to maintain the current organization, as we believe the existing structure best preserves the conceptual flow and logical connections between ideas.
Comment 5: We thank the reviewer for noting this omission. Appropriate citations have now been added to support the statement: “Although its complete in vivo function remains unclear, recent findings suggest that adipsin also supports insulin secretion by pancreatic β cells and protects these cells from apoptosis.”
Comment 6: We appreciate this suggestion and have now added a clear definition in the text:
“Type 2 immunity mediates defense against helminths, contributes to allergic inflammation, and supports tissue homeostasis and repair. It involves innate cells such as group 2 innate lymphoid cells (ILC2s) and eosinophils, as well as adaptive type 2 helper T (Th2) cells, all characterized by the production of interleukins IL-4, IL-5, and IL-13.”
Comment 7: We thank the reviewer for this important comment. The conclusion has now been revised.